# Pelvic Actinomycosis and Diagnostic Complexity: Case Report with Literature Review

**DOI:** 10.3390/healthcare13050485

**Published:** 2025-02-24

**Authors:** Stefano Restaino, Benedetta Gomba, Cecilia Zero, Guglielmo Stabile, Carlo Ronsini, Luigi Della Corte, Stefano Cianci, Federica Perelli, Ilaria Piacenti, Lorenza Driul, Giovanni Terrosu, Filippo Bordin, Cristina Taliento, Chiara Ripepi, Pantaleo Greco, Martina Arcieri, Giuseppe Vizzielli

**Affiliations:** 1Department of Obstetrics and Gynaecology, ASUFC, Ospedale Santa Maria della Misericordia, 33100 Udine, Italy; lorenza.driul@asufc.sanita.fvg.it (L.D.); martina.arcieri@asufc.sanita.fvg.it (M.A.); giuseppe.vizzielli@asufc.sanita.fvg.it (G.V.); 2PhD School in Biomedical Sciences, Gender Medicine, Child and Women Health, University of Sassari, 07100 Sassari, Italy; 3Obstetrics and Gynaecology, Department of Medical Area (DAME), University of Udine, 33100 Udine, Italy; gomba.benedetta@spes.uniud.it (B.G.); zero.cecilia@spes.uniud.it (C.Z.); bordin.filippo@spes.uniud.it (F.B.); 4Department of Medical and Surgical Sciences, Institute of Obstetrics and Gynaecology, University of Foggia, 71122 Foggia, Italy; guglielmost@gmail.com; 5Department of Woman, Child and General and Specialized Surgery, University of Campania “Luigi Vanvitelli”, 81100 Naples, Italy; carlo.ronsini@unicampania.it; 6Department of Neuroscience, Reproductive Sciences and Dentistry, School of Medicine, University of Naples Federico II, 80138 Naples, Italy; luigi.dellacorte@unina.it; 7Unit of Gynecology and Obstetrics, Department of Human Pathology of Adult and Childhood “G. Barresi”, University of Messina, 98122 Messina, Italy; stefano.cianci@unime.it; 8Azienda USL Toscana Centro, Gynecology and Obstetric Department, Santa Maria Annunziata Hospital, 50012 Florence, Italy; federica.perelli@gmail.com; 9Department of Obstetrics and Gynecology, Santa Maria Della Stella Hospital, 05018 Orvieto, Italy; i.piacenti@aospterni.it; 10General Surgery Clinic and Liver Transplant Center, University Hospital of Udine, 33100 Udine, Italy; giovanni.terrosu@uniud.it; 11Department of Obstetrics and Gynecology, University Hospital Ferrara, 44124 Ferrara, Italy; cristinataliento1@gmail.com; 12UOC Clinica Ostetrica e Ginecologica, Department of Medicine, Surgery and Health Sciences, University of Trieste, 34127 Trieste, Italy; chiara.ripepi@studenti.units.it; 13Department of Medical Sciences, Obstetrics and Gynecology Unit, University of Ferrara, 44121 Ferrara, Italy; pantaleo.greco@unife.it

**Keywords:** actinomyces, pelvic actinomycosis, pelvic inflammatory disease, intrauterine device

## Abstract

**Introduction**: Actinomyces are commensal bacteria that colonize various sites in the human body. Under certain predisposing conditions, they can proliferate and cause granulomatous inflammation in different tissues. Pelvic actinomycosis is an extremely rare condition; its significance lies in the fact that it can be easily misdiagnosed as gynecological malignancy based on clinical manifestations and imaging findings. According to the literature, this type of inflammatory disease is a common mimic of ovarian cancer. **Objective and methods**: This paper aims to provide a comprehensive overview of the characteristics of pelvic actinomycosis through a literature review and the description of two cases collected between 2023 and 2024 at Azienda Ospedaliera Universitaria Santa Maria della Misericordia in Udine Italy.

## 1. Introduction

Actinomyces are obligate anaerobic Gram-positive bacilli that exist as commensals within the human oral cavity, urogenital tract, and gastrointestinal tract [1]. Despite the presence of peripheral hyphae, Actinomyces is classified as true bacteria due to the absence of mitochondria and a nuclear membrane [2]. These organisms may become pathogenic when mucosal barriers are compromised, allowing for invasion. Actinomycosis is characterized by suppurative granulomatous inflammation, resulting in the formation of abscesses accompanied by draining sinus tracts or, in certain instances, the presence of fistulas [1].

Actinomycosis, as noted by Koteeswaran [1], can present with various localizations. For instance, cervicofacial presentation is the most common, affecting the skin and soft tissues in the jaw and presenting as painless masses with multiple abscesses. It can also involve the thorax, displaying symptoms similar to those of chronic lung infection and cancer. In the abdomen, it often affects the appendix and colon with nonspecific symptoms; in such cases, abscesses may form abdominal fistulas. Skin involvement is characterized by large painless soft tissue swellings that may be destructive and extend to the bone.

Alternatively, the second prevalent site of actinomycosis is the urogenital tract. Pelvic actinomycosis frequently arises in women who have a history of intrauterine device (IUD) insertion. Patients may present with nonspecific clinical manifestations, such as pelvic pain and vaginal discharge; however, fever is not typically observed as a symptom. Based on ultrasound examination, these findings are often misinterpreted as gynecological tumors; imaging results commonly reveal the presence of tubo-ovarian masses that may develop during the progression of the condition time.

Actinomyces infections are rare conditions; the annual incidence rate of actinomycosis is reported to be approximately about 0.3 cases per million people per year [3]. Pelvic actinomycosis represents 5% of all actinomycosis cases [4].

The diagnosis of actinomycosis is made post-operatively through histological evaluation. In patients exhibiting this clinical presentation alongside ultrasound results, surgery is performed as the initial treatment. Histopathological analysis may reveal suppurative granulomatous inflammation characterized by granulation tissue and colonies of Actinomyces bacteria, which can be visualized through immunofluorescence techniques.

Actinomycetes show sensitivity to penicillin G and beta-lactam antibiotics. Pharmacological treatment typically lasts for 6 to 12 months and generally complements surgical intervention in more complex cases.

## 2. Case Presentation

We present two cases from the Department of Gynecology and Obstetrics at Azienda Ospedaliera Universitaria Santa Maria della Misericordia in Udine, Italy, between 2023 and 2024. Over a period of two years, two cases were documented at Azienda Ospedaliera Universitaria Santa Maria della Misericordia in Udine, thereby underscoring the rarity of this pathology.

### 2.1. Case #1

The first patient, a 48-year-old woman, came to the general Emergency Room in March 2023. She had been experiencing abdominal pain, distention, and intestinal obstruction a few days before her visit. Importantly, she exhibited no signs of fever and had no notable medical history. A CT scan showed an enlarged right adnex with multiple fluid-filled cysts (Figure 1). The left side of the uterus appeared thickened and was in continuity with the left bladder wall (Figure 2). Moreover, the colonic wall displayed thickening and hyperemia. The abdominopelvic area contained heterogeneous material with fluid pockets and signs of peritoneal fluid effusion. Consequently, the patient was referred for a gynecological evaluation. Clinical findings indicated abdominal distension accompanied by pain upon palpation. A bimanual examination uncovered indeterminate solid tissue. The ultrasound revealed a thin, linear endometrium and an enlarged right ovary, while the left ovary was significantly adhered to the sigmoid colon. Laboratory results showed an elevated white blood cell count (15.02 × 10^3^/µL) and a higher C-reactive protein level (CRP 40.49 mg/L). Due to the high suspicion of gynecological malignancy, the patient underwent surgical procedures including laparotomic hysterectomy, bilateral tubo-oophorectomy, sigmoid resection with ileostomy, removal of the round ligament, omentectomy, appendectomy, and peritoneal biopsies. Intraoperative observations noted substantial adhesions involving the uterus, ovaries, and the sigmoid–rectal tract, along with mild ascitic effusion. An intraoperative histological assessment identified Actinomyces colonies. Despite this, the extensive disease burden suggested that demolitive surgery was warranted. Histological examination of the specimen diagnosed pelvic inflammatory disease featuring a tubo-ovarian abscess, revealing necrotic tissue with inflammatory characteristics, significant inflammation, and Actinomyces colonies mainly in the round ligament and right ovary. As a complication, the patient required a bilateral pyelostomy because of a left ureteral fistula linked to pelvic effusion stemming from past inflammation and vesicoureteral reflux. Nine months later, she underwent ano–rectal fistula repair. She received three months of antibiotic therapy with Amoxicillin and Doxycycline, followed by an additional eleven months of Amoxicillin, depending on complications after surgery. Fourteen months post-initial procedure, an X-ray with barium enema/CT scan was performed, indicating significant remission of complications. Currently, she is asymptomatic and continues to attend follow-up appointments monitoring.

### 2.2. Case #2

The second patient is a 54-year-old woman who was referred for gynecological evaluation by her private physician. She presented with vaginal bleeding that persisted for one month and had a concurrent intrauterine device (IUD) in place. Her medical history did not reveal any significant pathologies, aside from the recent discovery of a kidney cyst for which she underwent a Uro-CT scan one month prior. The CT scan indicated the presence of a pelvic mass of adnexal origin with suspicious characteristics and pelvic lymphadenopathy. Clinical gynecological evaluation did not demonstrate abnormal findings; the abdomen was manageable with no elicited pain. Ultrasound examination revealed an irregular uterine profile due to the presence of leiomyomas; the right ovary appeared normal, while the left ovary was completely occupied by a solid mass characterized by irregular borders, heterogeneous content with cystic areas, and a hypo-anechoic image. This mass exhibited poor vascularization on Doppler imaging and showed firm adhesion to the posterior lateral wall of the uterus. Laboratory tests indicated leukocytosis with neutrophilia (WBC 14.75 × 10^3^/µL, neutrophils 75%) and elevated C-reactive protein (CRP 149.71 mg/L). Tumor markers returned negative results. The IUD was removed, and the patient was forwarded for immediate CT evaluation. The imaging confirmed the presence of a left adnexal mass containing cystic or necrotic components, in continuity with the left uterine wall. Additionally, the endometrium displayed inhomogeneity. The mass was surrounded by moderate tissue effusion and showed adherence to the left ureter. The sigmoid colon exhibited thickened walls. Lymphadenopathy was noted in various pelvic locations. The patient underwent surgical treatment involving laparotomic en-bloc resection of the uterus, bilateral adnexa, sigmoid–rectal segment, colonic anastomosis, ureteral resection with reimplantation and stent placement, lymphadenectomy, omentectomy, and resection of the falciform hepatic ligament. Intraoperative frozen section histological examination indicated a diagnosis of acute inflammation. The definitive histological evaluation revealed the presence of Actinomyces colonies within the context of diffuse acute inflammation with abscess formation and extensive fibrotic reaction. Three months post-surgery, a follow-up CT scan indicated disease remission. The patient was prescribed a course of antibiotic therapy with Amoxicillin and Doxycycline for a duration of five months, and as of this writing, she continues to receive Amoxicillin therapy. Currently, the clinical situation is stable, the patient exhibits no residual symptoms, and laboratory findings indicate an optimal response to treatment.

## 3. Literature Review

Pelvic actinomycosis, while an exceptionally rare condition, has been documented in the literature regarding its clinical presentation, infection pathogenesis, and the crucial need for differential diagnosis with gynecological cancers. We conducted an extensive search of PubMed and Scopus from January 1990 to November 2024 to locate case reports or series detailing clinical instances of pelvic actinomycosis (see Table 1). The literature indicates that pelvic actinomycosis accounts for 3% of all human actinomycosis cases, making it the second most common infection site after the cervicofacial region. In 1976, it was deemed one of the rarest diseases, with only 300 reported cases at that time. As pointed out by Evans, the prevalence of anaerobic culture and histological analyses for surgically treated pelvic abscesses was quite low, leading to frequent underdiagnosis of the infection. Among various Actinomyces species, Israelii is frequently cited as the primary causative agent in humans, according to numerous studies [5].

Discussion: We analyzed findings from last 15 years. They included both case reports and literature reviews. The main symptoms were abdominal pain and fever, with two cases of nonspecific presentation and one case of asymptomatic presentation. Association with an IUD is present in the majority of articles reviewed. The literature shows concordant results also analyzing laboratory testing and the use of antibiotic therapy; for the first topic, the main evidence is about the elevation of WBC and CRP. Antibiotic therapy was performed in the majority of cases, suggesting that it is suitable and efficient for the treatment of this pathology.

Actinomycosis is often seen in women using IUDs, a connection first noted in 1928 [13]. This link is now thoroughly documented [14]. In his 2022 paper, Koteeswaran reiterates this, concluding that the use of IUDs heightens the risk of pelvic actinomycosis. He details the pathogenesis of pelvic Actinomyces infections, mentioning that Actinomyces-like organisms inhabit the cervix, vulva, and perineum. The insertion of an IUD can traumatize the epithelium, eroding the mucosal barrier and making it easier for Actinomyces to invade. Ferjaoui et al. [7] clearly outline that 75% of patients in their analysis had an IUD history, averaging 8.44 years of use years.

Additional risk factors for acquiring Actinomyces infections include existing infections from bacterial vaginosis pathogens, which create an anaerobic environment conducive to Actinomyces growth [8]. Many researchers argue that infections in the reproductive tract caused by bacterial vaginosis pathogens, coupled with a decrease in normal vaginal lactobacilli, lead to a higher pH and reduced acidity, ultimately promoting the growth of Actinomyces proliferation. Koteeswaran clearly explained that the “Splendore-Hoeppli” phenomenon—a dark eosinophilic rim surrounding Actinomyces colonies, formed due to immune complex deposits and cell debris [9]—enables bacteria to evade the immune system and suppress its response phagocytosis. We should also take into account previous surgeries, such as those for appendicitis and colonic diverticulitis, as risk factors for gastrointestinal actinomycosis. As Valour [2] suggests, we must not overlook immunosuppressive conditions, like HIV infection and organ transplantation, as well as malignancies, which are additional risk factors.

Regarding clinical manifestations, many authors agree that the symptoms of pelvic actinomycosis are nonspecific. Some literature highlights cases of women suffering from lower abdominal pain and loss of appetite lasting three months, without fever or vaginal discharge; typically, this pain appears intermittently and is not linked to menstrual cycles [1]. According to Ferajoui’s study [7], pelvic pain was the most frequently reported complaint, followed by fever and vaginal bleeding. Dhillon et al. [10] describe a patient lacking specific symptoms but experiencing episodes of fever, night sweats, anorexia, and unintentional weight loss; she also reported urinary frequency without dysuria and irregular bowel habits. In this case, subsequent investigations revealed that vague pelvic discomfort was reported seven months later.

Clinical examinations indicated consistent findings across various literature sources. As reported by Ferajoui et al. and Koteeswaran, pelvic masses can be identified through palpation, with the masses characterized as both firm and tender. Typically, an ultrasound examination is conducted, confirming these masses’ presence, revealing a solid component that raises suspicion for neoplastic disease. The masses can also be solid or cystic, featuring a thick, net-like wall resembling ovarian malignancies [6]. In the context of addiction, lymphadenopathy is observed in 50% of patients diagnosed with pelvic actinomycosis, thereby supporting the hypothetical diagnosis of malignancy. Most cases of actinomycosis associated with intrauterine device (IUD) insertion result in the formation of tubo-ovarian abscesses, which are likely to be misidentified as ovarian tumors. In certain instances, CT scans were conducted and revealed significant inflammation in the pelvic region. An additional finding involved the development of hydronephrosis, which led clinicians to infer a malignant ureteric obstruction [10].

Regarding laboratory testing, numerous authors note that patients often present with elevated WBC and CRP levels. Specifically, blood tests typically reveal leukocytosis and neutrophilia. While tumor marker CA 125 levels may show a slight increase, it is important to remember that these levels can also rise in cases of PID and endometriosis [15].

Literature indicates that surgical treatment has been the primary approach to date. In the study by Ferjaoui, 8 out of 12 patients received surgical interventions, predominantly total hysterectomy with bilateral salpingo-oophorectomy; one case also required bowel resection to treat a tubo-intestinal fistula. The primary intraoperative observation was a tubo-ovarian abscess, which in some instances was either bilateral or extended to the appendix or omentum. Surgery is often essential for a definitive correction diagnosis. Dhillon states that microbiological cultures are negative in 76% of instances, leading to the conclusion that the preoperative diagnosis derived from fluid culture provides accurate diagnoses only in 10% of cases [10]. At present, the definitive method for diagnosing actinomycosis is through histopathological examination coupled with bacterial culture obtained from a tissue biopsy [1].

According to Koteeswaran [1], histological examination of Actinomyces colonies demonstrate typical features. Hematoxylin and eosin staining shows clumps of filamentous bacteria; these display a dark staining eosinophilic rim at the periphery, referred to as the “Splendore-Hoeppli” phenomenon, caused by immune complex deposits and cell debris. It is also typical to observe suppurative inflammation, with the presence of numerous polymorphonuclear neutrophils.

Surgical treatment typically involves demolitive procedures, with total hysterectomy and salpingo-oophorectomy being the preferred options. In certain cases, surgical removal of infected tissue may be required to drain abscesses affecting the bowel, ureters, or other organs [10]. Purely surgical approaches have been found to offer minimal benefit [16]. In addition, recent findings from Han Y. et al. [17] affirm that due to the effectiveness of antibiotic treatment, surgical intervention is limited to excising necrotic tissue or draining abscesses. Actinomyces show sensitivity to Penicillin G and beta-lactams; additionally, Clindamycin, Macrolides, Tetracycline, and Doxycycline are effective alternatives for patients with penicillin allergies or those exhibiting antibiotic resistance. The selection of the treatment regimen considers the affected site, any coexisting pathogens, and the specific strain of infection actinomycosis. The preferred therapy regimen is Penicillin G, which is administered as an intravenous infusion for 2–6 weeks and then orally for a total of 6–12 months [17]. As for PID caused by other agents, early administration of antibiotics is necessary to reduce the risk of long-term sequelae (chronic pelvic pain, tubal infertility, ectopic pregnancy) [18]. The intrauterine device (IUD) should be removed if it is present. It is recommended to conduct regular follow-up assessments, both clinically and through radiological means [12].

It has been noted that pelvic actinomycosis is often misdiagnosed as an ovarian tumor. Koteeswaran [1] assists in examining the differential diagnosis. Both primary ovarian cancer and metastasis, alongside colorectal cancers, should be acknowledged as potential considerations. Furthermore, pelvic inflammatory disease also emerges as a plausible diagnosis, and chronic infections, particularly in the absence of fever—such as tuberculosis—must be contemplated, particularly in patients hailing from regions with a high prevalence of such conditions [6]. Endometriosis affecting the ovary constitutes a source of ovarian masses and abdominal pain but typically aligns with menstruation and may also result in infertility. In instances of unilateral pain accompanied by missed menstrual periods, ectopic pregnancy should be considered. Additionally, gastrointestinal issues, such as diverticulitis, acute appendicitis, or inflammatory bowel disease, should be included in the differential diagnosis list.

## 4. Discussion

The analyzed cases align with the existing literature. Initially, both patients were referred for gynecological evaluation due to findings of a pelvic mass. The first patient exhibited more pronounced symptoms, like pain and intestinal obstruction, whereas the second patient experienced less significant manifestations. Therefore, clinical presentations vary, and symptoms are typically nonspecific.

Considering the imaging findings, the pelvic masses observed in both cases presented characteristics indicative of a potential malignancy. The masses were primarily associated with adnexal structures, exhibiting either cystic or solid contents and forming adhesions with adjacent anatomical entities. Although the ultrasound examination may have yielded nonspecific results, the CT scan also displayed images suggestive of gynecological malignancy. It would be improbable to suspect a benign pathology based solely on clinical examination and imaging findings, corroborating the existing literature.

Surgical intervention was conducted in both cases. The procedure involved a demolitive approach through major surgery, which aimed to excise all affected tissues. Nevertheless, it became essential to establish a definitive diagnosis. Histological examinations revealed the presence of both colonies of Actinomyces alongside diffuse, acute suppurative inflammation characterized by abscesses and areas of necrosis. In both instances, surgical treatment was imperative to eradicate the pathology due to the advanced stage of the disease. Prolonged antibiotic therapy was administered, taking into account potential complications and the severity of the illness remission.

In both instances, the initial diagnostic suspicion, based on imaging findings and clinical symptoms, was for ovarian cancer. Additionally, our department is an Oncological Center of Expertise, certified by the European Society of Gynecological Oncology (ESGO); the imaging findings were considerably indicative of ovarian cancer, even when juxtaposed with those typically encountered in our practice. It is also important to note that none of the patients exhibited vaginal bleeding or other vaginal abnormalities discharge. Certain authors, including Westhoff et al. [11], assert that it is possible, in specific cases, to detect Actinomyces-like organisms in Pap test smears. This notion may warrant future consideration for the implementation of such examinations, potentially in conjunction with vaginal swab tests, prior to proceeding with surgical interventions. Additionally, it is imperative to evaluate whether the patient’s clinical presentation, symptoms, and overall condition permit and allow sufficient time to await microbiological results while contemplating differential diagnoses.

We would like to direct your attention to two pertinent aspects. The first pertains to the initial patient, who experienced complications subsequent to the previous surgical procedure. These complications included vesicoureteral reflux and anal–rectal fistula, which necessitated bilateral pyelostomy placement and subsequent fistula repair, respectively. Additionally, antibiotic therapy was also administered longer. In analyzing the overall situation, we may consider that the initial stage of the disease, when the patient came to our attention, was advanced and compromised. The abdomen already exhibited distention, and the bowel was obstructed by gas and feces, indicating an advanced illness at the beginning of our evaluation. We assume that this type of surgery exposes patients to a higher risk of complications. Firstly, not only the uterus and adnexa, but also various other organs, may be impacted by inflammation, and should be considered for surgical intervention. Secondly, surgeons must contend with inflammatory tissues that can pose management challenges and may necessitate an extended healing period.

Another important aspect pertains to the fact that the second patient had an intrauterine device (IUD) at the time of diagnosis. Given the prevalent assertion in the literature that IUD usage increases the risk of pelvic actinomycosis, it is prudent to consider this pathology in this patient. This suspicion should be corroborated by the laboratory tests that revealed an increase in white blood cell (WBC) count, particularly neutrophilia, as well as an elevation in C-reactive protein (CRP); additionally, tumor markers were found to be negative. Nonetheless, even if a benign condition was suspected, surgical intervention should have been deemed necessary due to the severity of the illness as evidenced by imaging studies, thus facilitating a definitive diagnosis in accordance with established literature states.

When comparing the two cases, it is noticeable how the postoperative trend is different. In the first case, we identified more complications which can lead to a need for longer antibiotic therapy. The second patient, who presented higher levels of blood CRP at the beginning, experience more benefits from pharmacological therapy. In our analysis, a reason for this disparity should be due to the different presentations and symptoms, which were more severe in the first case, justifying a more complicated course of pathology and its remission. On the other hand, the patient in the second case had an IUD at the time of the diagnosis, which is a risk factor for actinomycosis development. IUD removal and antibiotic therapy should have been an important step towards the resolution of this pathology, which requires surgical intervention in any case in order to obtain a definitive diagnosis

## 5. Conclusions

Pelvic actinomycosis represents an exceedingly rare condition. Actinomycosis exhibits a non-specific presentation with varying clinical findings ranging from moderate to severe; the symptoms overlap with numerous other conditions, complicating the differential diagnosis. Imaging findings can be ambiguous and may easily lead to the misdiagnosis of malignancy. Based on both the literature and our experiences, patients typically undergo surgery. The surgical approach remains the primary treatment for pelvic issues due to the necessity of histopathological examination as the gold standard for diagnosis and to rule out malignant actinomycosis; it is usually supported by a targeted and long-term antibiotic therapy. A compelling future direction could involve focused research on diagnostic tools that offer clearer indications for suspecting pelvic actinomycosis. A key objective may be to enhance diagnostic techniques to prevent invasive surgery, particularly for patients wishing to preserve their health and fertility.

## Figures and Tables

**Figure 1 healthcare-13-00485-f001:**
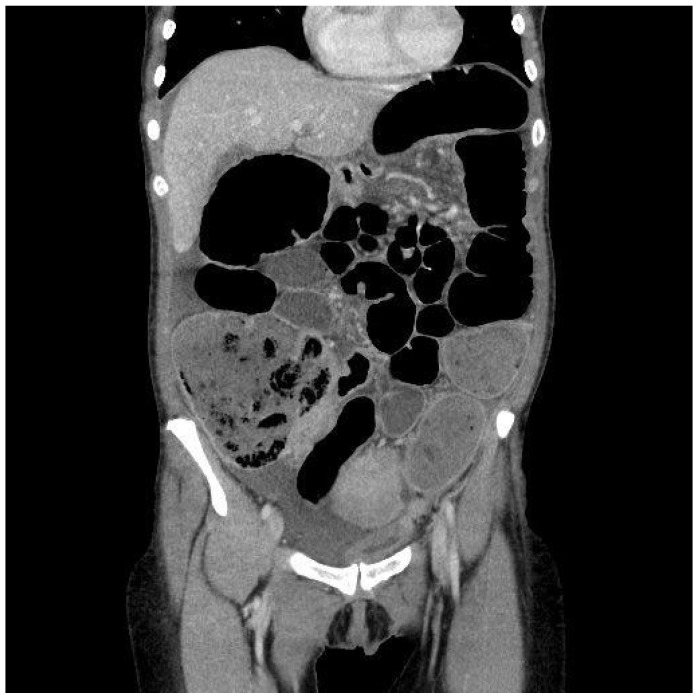
Frontal CT scan. It is possible to appreciate the right adnex mass with disomogeneous solid and cystic content. This is a common aspect between Actinomycosis and ovarian cancer. Bowel dilatation should make clinicians suspect neoplastic sub-occlusion and omentum suffusion should mimicry neoplasm dissemination, leading to a misdiagnosis.

**Figure 2 healthcare-13-00485-f002:**
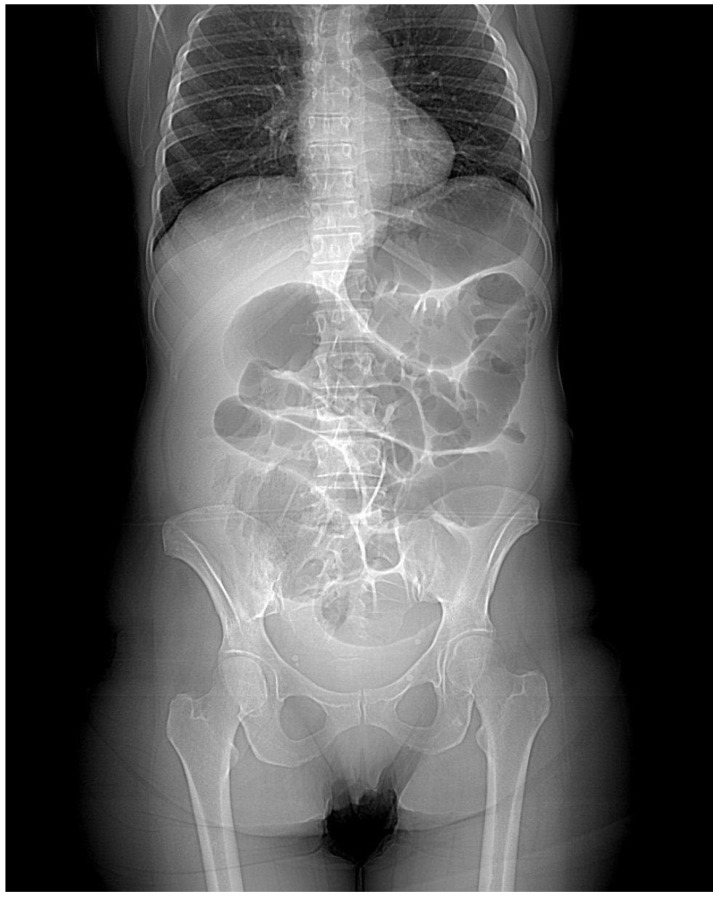
Frontal CT scan. In this case, the main aspect is diffuse and consistent bowel dilatation, which can be a common aspect in stenotizing cancer infiltration. However, for a more detailed look, it is possible to appreciate bowel thickened walls and phlogistic effusion, which can make clinicians suspect an inflammatory disease.

**Table 1 healthcare-13-00485-t001:** Data of the systematic review.

Author	Year of Publication	Tipe of Article	Patients’ Symptoms	IUD	Laboratory	Imaging Performed	Treatment	Histology	Antibiotic Therapy
Koteeswaran, R. et al. [1]	2022	Case report	Abdominal pain, lack of appetite	No	Elevated WBC, CRP, CA 125	Ultrasound, CT scan	Demolitive surgery	Actinomyces colonies, abscess	Yes
Valour, F. et al. [2]	2014	Review and case resport	Acute peritonitis with fever	Yes	Elevated WBC, CRP, CA 125	Ultrasound, CT scan	Open surgical resection	Actinomycosis phlogosis	Recommended
Saramago, S.M. et al. [6]	2019	Case report	None	Yes	Anemia	CT scan, pelvic MR	Laparotomic surgery	Active chronic inflammation and abscess formation.	Yes
Mbarki, G. et al. [4]	2016	Retrospective study	Pelvic pain, fever	87% yes	Elevated WBC and CRP	Ultrasound, CT scan	Surgery	Suppurative inflammation	Yes
Sharma, S. et al. [5]	2021	Review	Lower abdominal pain, constipation, vaginal discharge. Fever is rare	Common	Leukocytosis, elevated neutrophils. CRP and ESR elevated	CT scan	Antibiotics and surgery	/	Yes
Ferjaoui, M.A. et al. [7]	2021	Retrospective study	Pain 66%, fever 33%, vaginal bleeding 25%	75% yes	Elevated WBC 58%, elevated CRP 58%, anemia 83%, elevated CA 125 66%	TSV ultrasound	75% surgery	Chronic inflammation, giant multinucleated cell with Actinomyces colonies	Yes
García-García, A. et al. [8]	2017	Review	Does not produce characteristic disease signs or symptoms	Frequent	Leucocytosis, erythropenia, high ESR and CRP	Ultrasound, CT scan, MR, X Ray	Surgery	Inflammation of suppurative/granulomatous nature, connective proliferation, sulfur granules	Yes
Gopinath, D. et al. [9]	2018	Scientific paper	/	/	/	/	/	/	/
Dhillon Ajit Kaur et al. [10]	2015	Case report	Abdominal pain, fever, night sweats, anorexia, and weight loss	Yes	Elevated WBC, CRP, Ca 125, lack of hemoglobin	Ultrasound, Ct scan, X-ray	Demolitive surgery	Abscesses and suppurative phlogosis	Yes
Westhoff, C. et al. [11]	2007	Review	Nonspecific	7% of women using an IUD may have finding of Actinomyces on a Pap test	Nonspecific	Ultrasound, CT scan, nonspecific	Usually surgical	/	Yes
Wong, W.K. et al. [12]	2011	Review	Nonspecific symptoms (fever, weight loss, abdominal pain).	Frequently	Anemia, leucocytosis, raised ESR, and raised CRP values	Ultrasound, CT, and MR are nonspecific	Should be considered	Gram and filamentous organisms and sulfur granules are supportive of a diagnosis of actinomycosis	Recommended

## Data Availability

The data presented in this study are available on request from the corresponding author.

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
