# Peer review of "Pelvic Actinomycosis and Diagnostic Complexity: Case Report with Literature Review"

_healthcare, 2025, doi:10.3390/healthcare13050485_

Round 1

Reviewer 1 Report

Comments and Suggestions for Authors

1. Clarity and Cohesion

The manuscript presents a clear and logical structure, addressing the diagnostic complexity of pelvic actinomycosis through the description of two clinical cases and a literature review. However, some sections lack a smoother connection between the presented information. It is recommended to revise the transition between the clinical cases and the literature review to enhance readability and coherence.

Additionally, there is repetition of information throughout the text, particularly regarding the diagnostic challenge of pelvic actinomycosis. For instance, the mention of differential diagnosis with gynecological neoplasms occurs in several passages and could be consolidated into a single paragraph in the introduction or discussion.

2. Introduction

The introduction provides an adequate overview of Actinomyces pathogenicity, highlighting its clinical presentation and the difficulty in differentiating it from gynecological cancer. However, some information could be better organized to improve the text’s flow.

Suggestions for improvement:

 The introduction should be more concise in defining Actinomyces and its pathogenesis, avoiding redundancies that are later repeated in the literature review.

 Including more up-to-date epidemiological data on the incidence of pelvic actinomycosis, if available, would better contextualize the relevance of the study.

 A brief justification for the selection of the presented cases would strengthen the study’s scope.

3. Clinical Case Reports

The description of the two clinical cases is detailed and follows a logical flow, providing relevant information about symptoms, imaging tests, treatment, and clinical progression. However, certain aspects could be improved:

 The first case is described with an adequate level of detail, but some sentences are long and complex, which may compromise clarity. Revising sentence construction could facilitate readability.

 In the second case, the information about the intrauterine device (IUD) as a predisposing factor is relevant but could be better contextualized in the discussion to avoid redundancy.

 In both cases, the correlation between imaging findings and histopathological results could be discussed more explicitly, emphasizing the diagnostic challenges faced by the medical team.

4. Literature Review

The literature review is comprehensive and well-supported, with an extensive review of studies on pelvic actinomycosis. However, some limitations need to be addressed:

 The table of reviewed studies contains relevant information, but it lacks a critical summary discussing the findings comparatively. The table could be supplemented with a deeper analysis of the similarities and differences between the literature cases and those presented in the study.

 Some references are outdated and could be replaced with more recent articles, particularly those concerning the diagnosis and treatment of pelvic actinomycosis.

 The relationship between actinomycosis and IUD use was well explored, but the explanation of bacterial colonization could be clearer and more direct.

5. Discussion

The discussion section provides a satisfactory analysis of the reported cases, contextualizing them within the literature. However, certain points could be improved:

 The comparison between the two cases could be more explicit. For example, the first case had a more severe progression with postoperative complications, while the second case responded satisfactorily to treatment. Analyzing the possible reasons for this difference would enrich the discussion.

 The impact of late diagnosis on patient prognosis could be better emphasized, highlighting the need for greater attention to clinical and laboratory signs of actinomycosis.

 The discussion on the necessity of surgery in both cases is well-supported, but it would be interesting to comment on the possibility of less invasive treatments in patients diagnosed early.

6. Conclusion

The conclusion adequately synthesizes the study’s findings, highlighting the rarity of pelvic actinomycosis and the importance of differentiating it from gynecological neoplasms. However, some suggestions could improve this section:

 The need for new non-invasive diagnostic approaches could be better emphasized.

 The role of antibiotic therapy in disease management could be reinforced, indicating whether there is consensus in the literature regarding the ideal treatment duration.

 The conclusion could be slightly more concise, avoiding repetitions of information already discussed throughout the article.

Author Response

Thank you for your review. Your suggestions help us to improve the quality of this paper.

  1. Clarity and Cohesion

The manuscript presents a clear and logical structure, addressing the diagnostic complexity of pelvic actinomycosis through the description of two clinical cases and a literature review. However, some sections lack a smoother connection between the presented information. It is recommended to revise the transition between the clinical cases and the literature review to enhance readability and coherence.

Additionally, there is repetition of information throughout the text, particularly regarding the diagnostic challenge of pelvic actinomycosis. For instance, the mention of differential diagnosis with gynecological neoplasms occurs in several passages and could be consolidated into a single paragraph in the introduction or discussion.

Thank you for your detailed suggestion; we  reviewed the transition between case report and literature review. A differential diagnosis section is included in “literature review”

  1. Introduction

The introduction provides an adequate overview of Actinomyces pathogenicity, highlighting its clinical presentation and the difficulty in differentiating it from gynecological cancer. However, some information could be better organized to improve the text’s flow.

Suggestions for improvement:

  • The introduction should be more concise in defining Actinomyces and its pathogenesis, avoiding redundancies that are later repeated in the literature review.
  • Including more up-to-date epidemiological data on the incidence of pelvic actinomycosis, if available, would better contextualize the relevance of the study.
  • A brief justification for the selection of the presented cases would strengthen the study’s scope.

Thank you for your suggestion; we included more up-to-date epidemiological data on the incidence of pelvic actinomycosis. The justification for the selection of the presented cases is been added before case presentation.

  1. Clinical Case Reports

The description of the two clinical cases is detailed and follows a logical flow, providing relevant information about symptoms, imaging tests, treatment, and clinical progression. However, certain aspects could be improved:

  • The first case is described with an adequate level of detail, but some sentences are long and complex, which may compromise clarity. Revising sentence construction could facilitate readability.
  • In the second case, the information about the intrauterine device (IUD) as a predisposing factor is relevant but could be better contextualized in the discussion to avoid redundancy.
  • In both cases, the correlation between imaging findings and histopathological results could be discussed more explicitly, emphasizing the diagnostic challenges faced by the medical team.

Thank you for your comment; we reviewed sentence construction in presentation of case #1. The information about the intrauterine device (IUD) as a predisposing factor is contextualized in “literature review” section.

  1. Literature Review

The literature review is comprehensive and well-supported, with an extensive review of studies on pelvic actinomycosis. However, some limitations need to be addressed:

  • The table of reviewed studies contains relevant information, but it lacks a critical summary discussing the findings comparatively. The table could be supplemented with a deeper analysis of the similarities and differences between the literature cases and those presented in the study.
  • Some references are outdated and could be replaced with more recent articles, particularly those concerning the diagnosis and treatment of pelvic actinomycosis.
  • The relationship between actinomycosis and IUD use was well explored, but the explanation of bacterial colonization could be clearer and more direct.

Thank you for your suggestion; we added a critical summary discussing the findings comparatively in literature review table. We didn’t consider outdated references.

  1. Discussion

The discussion section provides a satisfactory analysis of the reported cases, contextualizing them within the literature. However, certain points could be improved:

  • The comparison between the two cases could be more explicit. For example, the first case had a more severe progression with postoperative complications, while the second case responded satisfactorily to treatment. Analyzing the possible reasons for this difference would enrich the discussion.
  • The impact of late diagnosis on patient prognosis could be better emphasized, highlighting the need for greater attention to clinical and laboratory signs of actinomycosis.
  • The discussion on the necessity of surgery in both cases is well-supported, but it would be interesting to comment on the possibility of less invasive treatments in patients diagnosed early.

Thank you for this suggestion we really appreciate; we gave more explicit comparison between the two cases analysing the possible reasons for prognostic difference. We had a more detailed focus on the possibility of less invasive treatments.

  1. Conclusion

The conclusion adequately synthesizes the study’s findings, highlighting the rarity of pelvic actinomycosis and the importance of differentiating it from gynecological neoplasms. However, some suggestions could improve this section:

  • The need for new non-invasive diagnostic approaches could be better emphasized.
  • The role of antibiotic therapy in disease management could be reinforced, indicating whether there is consensus in the literature regarding the ideal treatment duration.
  • The conclusion could be slightly more concise, avoiding repetitions of information already discussed throughout the article.

Thank you for suggestion;  we emphasized the role of antibiotic therapy in disease management ad we made conclusion more concise.

 We hope our reviews make our work able to meet your requirements.

Reviewer 2 Report

Comments and Suggestions for Authors

1. Inadequate Differential Diagnosis Discussion – Needs more depth on distinguishing pelvic actinomycosis from malignancy, tuberculosis, and chronic infections based on imaging and histology.

2. Overgeneralization of Surgical Approach – Surgery is not always required; the paper should clarify when surgery is necessary versus when prolonged antibiotic therapy alone is effective.

3.Lack of Detailed Imaging Interpretation – CT/MRI findings should be described more precisely, emphasizing key radiologic features that differentiate actinomycosis from other pelvic masses.

4. Results and Discussion Structure – These sections are not clearly separated, leading to confusion. Results should be structured logically, with better integration of tables. Add age of patients and type of antibiotic treatment.

5. Outdated Literature – Many references are older than 10 years despite the availability of more recent studies. The discussion should incorporate the latest evidence.

6. Histopathology Description – Needs more details on histological findings, including sulfur granules, granulomatous inflammation, and fibrosis, to enhance diagnostic clarity.

7. Case Presentation Clarity – The two cases should be presented more systematically, highlighting key diagnostic challenges and treatment decisions.

8. Antibiotic Therapy Details – Needs more specifics on antibiotic selection, duration, and outcomes, including cases where long-term antibiotic therapy prevents the need for surgery.

Comments on the Quality of English Language

The text contains long, unclear sentences and grammatical errors that need revision for improved readability and scientific accuracy.

Author Response

Thank you for your review. Your suggestions help us to improve the quality of this paper.

  1. Inadequate Differential Diagnosis Discussion – Needs more depth on distinguishing pelvic actinomycosis from malignancy, tuberculosis, and chronic infections based on imaging and histology.

Thank you for your suggestion. A Differential Diagnosis Discussion is provided in last paragraph of literature review. A more detailed description could seem redundant in our opinion

  1. Overgeneralization of Surgical Approach – Surgery is not always required; the paper should clarify when surgery is necessary versus when prolonged antibiotic therapy alone is effective.

Thank you for your suggestion. We clarified when surgery is necessary versus when prolonged antibiotic therapy alone is effective

  1. Lack of Detailed Imaging Interpretation – CT/MRI findings should be described more precisely, emphasizing key radiologic features that differentiate actinomycosis from other pelvic masses.

We appreciate you suggestion. We provide a more detailed description of CT scan images

  1. Results and Discussion Structure – These sections are not clearly separated, leading to confusion. Results should be structured logically, with better integration of tables. Add age of patients and type of antibiotic treatment.

Thank you for this note. We provided toa better integration of tables with a commented analysis.  Age of patients and type of antibiotic therapies are already expressed in text.

  1. Outdated Literature – Many references are older than 10 years despite the availability of more recent studies. The discussion should incorporate the latest evidence.

We appreciate your suggestion. We excluded from table of literature review papers older than 10 years.

  1. Histopathology Description – Needs more details on histological findings, including sulfur granules, granulomatous inflammation, and fibrosis, to enhance diagnostic clarity.

Thank you for this suggestion. We added a more detailed histopathology description from literature.

  1. Case Presentation Clarity – The two cases should be presented more systematically, highlighting key diagnostic challenges and treatment decisions.

Thank you for your suggestion. We highlighted differences between two cases in discussion section

  1. Antibiotic Therapy Details – Needs more specifics on antibiotic selection, duration, and outcomes, including cases where long-term antibiotic therapy prevents the need for surgery.

Thank you for your suggestion. We emphasized the regimen of suggested antibiotic therapy

You can find changes we have made highlighted in text. We hope our reviews make our work able to meet you requirements.

Reviewer 3 Report

Comments and Suggestions for Authors

The case studies provides valuable clinical insights, making the article highly informative for healthcare professionals dealing with rare pelvic infections.
The descriptions of imaging, surgical interventions, and histological findings add depth and educational value for clinical readers. Moreover, the emphasis on differential diagnosis and the need for histopathological confirmation highlights critical aspects of clinical decision-making.

Main remarks:
- While the outcomes for the first case are more detailed, the second case lacks follow-up information beyond a stable condition. A more comprehensive discussion on long-term prognosis would be beneficial.
- While case studies are inherently qualitative, incorporating epidemiological data or meta-analytical findings from the literature would give the paper a stronger evidence base.
- The systematic literature review contextualizes the findings and underscores the rarity and diagnostic complexity of pelvic actinomycosis. However, it could be improved by providing a more critical synthesis of the reviewed studies, focusing on diagnostic challenges, advancements and treatment outcomes.
- Discuss imaging findings more thoroughly, emphasizing the specific features that differentiate actinomycosis from ovarian malignancies.

Author Response

Thank you for your review. Your suggestions help us to improve the quality of this paper.

Main remarks:
- While the outcomes for the first case are more detailed, the second case lacks follow-up information beyond a stable condition. A more comprehensive discussion on long-term prognosis would be beneficial.

Thank you for your suggestion. The outcome discussion of case #2 is less detailed than case #1 due to the fact that follow up and treatment is continuing at the time of this writing.

- While case studies are inherently qualitative, incorporating epidemiological data or meta-analytical findings from the literature would give the paper a stronger evidence base.

Thank you for your note. We incorporate epidemiological data in introduction

- The systematic literature review contextualizes the findings and underscores the rarity and diagnostic complexity of pelvic actinomycosis. However, it could be improved by providing a more critical synthesis of the reviewed studies, focusing on diagnostic challenges, advancements and treatment outcomes.

Thank you for your suggestion. In our opinion, diagnostic challenges implications are widely outlined in sections “literature review” and “discussion”; the aim of the paper is to provide a summary of what is known about pelvic actinomycosis so far and to describe our experience with critical analysis.

- Discuss imaging findings more thoroughly, emphasizing the specific features that differentiate actinomycosis from ovarian malignancies.

We appreciate this suggestion. We reinforced description of imaging findings.

We hope our reviews make our work able to meet your requirements.

Round 2

Reviewer 1 Report

Comments and Suggestions for Authors

I would like to confirm that all the requested revisions have been properly addressed in the revised manuscript. I believe that, with these corrections, the work meets the journal’s criteria and can be considered for acceptance.

Reviewer 2 Report

Comments and Suggestions for Authors

I would like to thank the authors for the opportunity to review their interesting manuscript.
The authors made all the appropriate changes and I recommend the acceptance of their work in its present form.

Reviewer 3 Report

Comments and Suggestions for Authors

Comments and suggestions were taken into account. The manuscript improved significantly.